# Assessment of the Mechanical Properties of Selected PLA Filaments Used in the UAV Project

**DOI:** 10.3390/ma16031194

**Published:** 2023-01-30

**Authors:** Marcin Graba, Andrzej Grycz

**Affiliations:** Department of Metrology and Non-Conventional Manufacturing Methods, Faculty of Mechatronics and Mechanical Engineering, Kielce University of Technology, Aleja Tysiąclecia Państwa Polskiego 7, 25-314 Kielce, Poland

**Keywords:** design, 3D printing, static tensile test, mechanical properties, PLA, tensile diagram, FEM, topological optimization

## Abstract

The paper presents the evaluation of selected mechanical properties of PLA filaments used in the production of unmanned aerial vehicles (UAV). The manuscript covers a short description of the principles of pattern design and presents a shortened division of incremental manufacturing technologies. The authors present and argue the choice of material that can be used in the UAV drone design. This material has been subjected to experimental tests in order to assess selected mechanical properties that can be successfully adapted to CAD/CAM/CAE systems in order to conduct engineering analyses. The manuscript presents a discussion on the influence of the method of specimen production on selected mechanical properties, as well as the issue of errors made when measuring selected mechanical properties. In addition to the assessment of the mechanical properties of the selected filament, the manuscript indicates how to adapt the determined material constants to the FEM calculation model and presents the effectiveness of topological optimization in engineering design, which allows to reduce the weight of the drone frame by about at least 20% compared with the value originally assumed.

## 1. Introduction

The last decade in the world of engineering has been a continuous development of various techniques and technologies. It can be concluded that this is the result of Moore’s empirical law and the inclusion of the Internet in almost all areas of life. All this significantly facilitates the process of obtaining information and enables a significant increase in the level of complexity of vehicles and devices designed by engineers, to the extent that the use of dozens of specialized sensors and cameras in consumer goods becomes the norm [1,2]. Manufacturing technologies have also changed significantly over the last ten years. Three-dimensional printing, i.e., the additive manufacturing technology, is becoming widely used [3,4,5]. The past two years and the present show that the ever-evolving unmanned aerial vehicle (UAV) projects are also an immeasurable field of engineering in which to invest [6,7]. The combination of both the UAV design process and incremental manufacturing techniques allows to quickly create a fully functional prototype that can play a variety of roles [3,4,5,6,7]. However, the development of such a project requires the proper selection of materials in accordance with the art of engineering, from which the UAV (commonly known as a drone) will be made.

The most common civil UAVs are four-rotor structures, available in many stationary and online stores. Single-rotor rotorcraft with an extended tail rotor are less popular. Such structures, as shown by the market analysis, are cheaper than four-rotor models and are also characterized by a potentially lower unit production cost and smaller overall dimensions of the structure due to the slenderness of the hull [8,9,10,11]. It is also an excellent way to demonstrate the potential of incremental methods to fabricate and operate complex mechanisms such as a rotor control system [8]. An example of the original UAV project is shown in Figure 1 [8].

The UAV model presented in Figure 1, optimized in accordance with the industrial design, is characterized by a smooth connection of the plating elements, among which there are four main components: the nose of the vehicle, the main fuselage, the battery cover and the tail section. Such a solution will facilitate the production of elements. The presented model assumes a modular chassis connection and the possibility of using the vehicle without the plating. The appearance of some elements may change at the stage of construction, further optimization and the production itself, for which it is proposed to use additive manufacturing techniques [8,9,10,11].

All additive manufacturing technologies have one feature of the production process in common: the joining of material layers (with very low values of height in relation to the height and width of the layer) [3,4,5]. Layers in most of the technologies are flat, but there are experimental methods of applying three-dimensional layers, especially in FDM technology. This method has a better surface quality but is rarely used for geometric reasons [3,4,5,12,13,14,15].

Additive manufacturing methods are divided according to the method of applying and joining layers, as well as according to the state of aggregation and the structure of the feed material. Within this division, three main types of currently used technologies can be distinguished [3,4,5]:Technologies based on material extrusion: FDM, LMD, LDM;Technologies based on liquid input material: LCD, MLCD, SLA, DLP, HARP;Technologies based on charge material in the form of powder: SLS, DMLS, jet fusion.

An exemplary breakdown of additive manufacturing methods is shown in Figure 2 [8].

For the purposes of the implementation of the drone project presented above, it was decided to manufacture all structural elements with the fused deposition modeling (FDM) technology based on the extrusion of the material with the use of PLA polylactide as the input material, with the market name COLORFILL [8].

## 2. Experimental Research

In order to determine selected mechanical properties of the color fill filament, a uniaxial tensile test was performed in accordance with the European–British standard BS-EN ISO 527 [16] for plastics. Type 1BA specimens were used in the study, Figure 3 [16].

The specimens were made on the Creality Ender 3 device (Creality, Shenzhen, China) with built-in model 4.2.7 motherboard drivers, with a 0.4 mm diameter head and a modified airflow compared with the factory one (Figure 4). The firmware is based on the Marlin 2.0 structure (Marlin, Heteren, The Netherlands). The device operates in the FFF technology and is based on commands in the G language. The air temperature in the vicinity of the device during the production of specimens was 20 ± 3 ° C and its humidity was about 70%.

The G-code file was generated in Ultimaker Cura version 4.13.1 (Ultimaker, Utrecht, The Netherlands) for the built-in device profile and modified settings for material, structure and device movements. Grid fill type was assumed. It is a truss-like structure, consisting of two sets of intersecting walls at right angles at a constant distance from each other, creating empty areas of cuboids with a square base (Figure 5). This structure does not change along the Z axis.

As can be concluded in [12], in which an analogous test was performed for such a filling, the increase in strength is not fully linear and there are strength jumps for certain value fillings (Figure 6). Therefore, three different types of fillings were selected—30%, 60% and 90%—to minimize material consumption. In order to minimize the test errors, three specimens were made for each of the specimen and filling orientations. Table 1 shows the parameters that did not change during the specimen production process.

Similar studies on the determination of mechanical properties of specimens and components produced by 3D printing are described in many scientific publications and industry portals, among which, apart from [12], manuscripts [13,14,15] can be distinguished. In [12], the authors discussed the influence of infill pattern and density on material properties only for tensile testing in one direction and for one printing orientation [12]. Results presented by the authors show that the “Concentric” [12] infill pattern gives the highest ultimate tensile strength and yield strength (for example, if comparing with 100% infill, with a “Concentric” infill pattern of 90%, ultimate tensile strength is reduced by only 15%, while the yield stress is reduced by 20%). The 90% infill reduced the printing time by 30% [12]. In paper [13], the authors discuss the influence of process parameters on mechanical properties and printing time of FDM PLA printed parts using a design of experiment. The authors used the PLA material to fabricate nine specimens with varied infill density, layer thickness and part orientation [13]. The specimens were fabricated by the FDM process using a 3D printing machine [13]. A lot of mechanical properties were tested by the authors. The experiment was carried out to investigate the effects of infill density, layer thickness and part orientation on the mechanical properties (such as tensile, bending and compression strength) of the FDM technology [13]. The authors note that the relationship between mechanical properties and process parameters (such as infill density, layer thickness and part orientation) is a positive relationship [13]. Based on [13], it can be noted that the mechanical properties increase with increases in infill density and layer thickness [13]. In the next study [14], the authors present the material testing of 3D printed ABS and PLA specimens to guide mechanical design, while paper [15] discusses the effect of filling patterns on the tensile, flexural and mechanical properties of FDM 3D-printed products. In [14], a set of monotonic tensile tests was performed on 3D printed plastics following ASTM standards [14]. The experiment in [14] tested a total of 13 “dog bone” test specimens, where the infill percentage, infill geometry, load orientation, strain rate and the material were varied [14]. The results obtained by the authors present that the ultimate tensile strength decreases as the infill percentage decreases and that the hexagonal pattern infill geometry was stronger and stiffer than the rectilinear infill [14]. The authors’ finite element method results indicate that the rectilinear infill showed less deformation than the hexagonal infill when the same load was applied [14]. In [15], the experimental study investigates the effect of filling patterns on tensile and flexural strength and modulus of the parts printed via fused deposition modeling [15]. The authors discussed different kinds of filling and some infill densities. The presented research indicates that a specific concentric pattern yields the most desirable tensile and flexural tensile properties at all filling densities [15]. Based on our own analysis and results presented by [12,13,14,15], it is appropriate to study the influence of 3D printing parameters on the selected mechanical properties of the COLORFILL filament, as mentioned above.

Figure 7 and Figure 8 show the orientation of the specimens during 3D printing, respectively. The first is the Ultimake Cura window and the second is a photo of physical specimens produced for the project described above. Table 2 presents a comparison of the filling of all options used in the research program for specimens printed in both vertical and horizontal orientations. With the increase in the degree of filling, the time of preparation of one specimen increases (which is an obvious conclusion), as well as the weight of the specimen (this is also an obvious conclusion). Comparing the mass of specimens with 30% and 90% filling, an increase in mass by about 30% can be noticed in relation to the specimen with less filling, while specimens printed in the horizontal orientation are characterized by a much greater mass than specimens printed in the vertical orientation, respectively, 39%, 52% and 48% more than specimens printed in the portrait orientation (Table 2). It should be remembered that these are values approximated by the program controlling the printing process.

After the specimens were printed, their physical tests were started. The study began by examining the mass of the specimens and comparing them with the values indicated by the slicer (Table 3). The program indications were also compared for one specimen printed horizontally and vertically with the support, where only the number of objects was changed. As can be seen, there are significant discrepancies for the three specimens printed simultaneously with respect to the actual weight. The relative error of slicer indications for specimens printed in series is on average 14.38%; for specimens printed individually, it is on average 13.86%. For specimens printed in the portrait orientation, the errors are twice as small as for specimens printed in the landscape orientation. Based on the results, it can be concluded that the program used estimates the mass value more accurately for elements with greater height than width and length. Such a discrepancy may be influenced by the inaccuracy of the algorithm for calculating the mass. The total solid density (together with the air under the shell) of the specimen was also determined, assuming the specimen volume V = 6941.38 mm^3^ indicated by the Solid Works 2021 program [17], in which the specimen models were created. These are the values that allow you to estimate the mass of the elements. It should be noted that, in reality, the density values may be slightly higher due to the gaps between the layers in the outer surface not included in the specimen model.

## 3. Measurement Results of Selected Mechanical Properties

The 18 uniaxial tensile tests were performed. The tests were carried out in accordance with the aforementioned BS EN ISO 527-4 procedure [16]. The specimens were mounted on a Shimadzu Autograph AGX-V testing machine (Shimadzu, Kyoto, Japan). The ambient temperature during the test was 21 °C. The distance between the jaws was 115 mm (Figure 9). The average cross-sectional area of the specimen was determined by measuring the length and width in three places inside the datum. The tensile test speed was 2 mm/min. Time (s), force (N) and traverse (mm) signals were recorded. The tests were carried out until the specimen cracked. Most of the specimens cracked at the boundary of the measuring area.

The conventional yield strength R_0_._2%_ was determined for the curves in which they performed. The modulus of linear elasticity was determined in accordance with the standard, dividing the increase in stress by the increase in strain on the section from $\varepsilon_{0.05\%}$ to $\varepsilon_{0.25\%}$:(1)$$E=\frac{\Delta \sigma }{\Delta \varepsilon },$$
where $\Delta \varepsilon =\varepsilon_{0.25\%}-\varepsilon_{0.05\%}$.

The test results are recorded in the tables below and presented in diagrams of normal stress versus normal strain. The breaking energy of the specimen was determined using the trapezoidal numerical integration method. All results are presented in Table 4, Table 5, Table 6, Table 7, Table 8 and Table 9 and in Figure 10 and Figure 11.

The conducted experimental studies (Table 4, Table 5, Table 6, Table 7, Table 8 and Table 9 and Figure 10 and Figure 11) show that the filling density significantly influences the mechanical parameters of printed objects. The greatest differences can be noticed in the case of the value of the linear elasticity modulus and the tensile strength limit. In the case of the first, for specimens printed from the same filament, the largest difference in the average values can be noticed for the COLORFILL 30% specimen in the landscape orientation, where the difference is approximately 0.73 GPa, which translates into a relative percentage error of 32.59% in relation to the maximum value for the 90% specimen. For the strength limit, it is the difference between 60% filling and 90% for the specimen in the horizontal orientation made of COLORFILL filament. The difference is 13.32 MPa, which translates into a relative error of 40.11%. The yield point did not show such significant discrepancies.

The orientation of the print significantly influences the behavior of the tension curve (Figure 10 and Figure 11). Vertical specimens were more likely to behave as a linearly brittle material (Figure 11). In this case, there was often no yield point. It resulted in the fact that the greatest deformations appeared at the boundaries of subsequent layers, which meant that the layers with a greater thickness than their ends did not deform to such a significant extent. This fact can also be noticed during the visual inspection of the breaking point (Figure 12). For vertical orientation, it has always been the boundary between two layers. For horizontal specimens, the break point was usually irregular, which means that stresses appeared along the filament threads themselves. The horizontal specimens behaved more vividly (Figure 12). However, a large irregularity can be noted here, because for some tests, after exceeding the ultimate strength denoted as R_m_, instead of breaking the specimen, the stresses decreased for a short period and the specimens remained unbroken. By analyzing the graphs and values (Table 4, Table 5, Table 6, Table 7, Table 8 and Table 9 and Figure 10 and Figure 11), it can be concluded that the material behaves in a rather unpredictable manner. There are large discrepancies between specimens in the same series for specimens stretched along the print direction. Most likely, this fact is influenced by the very characteristics of the specimen production process, where there are differences in the production time for subsequent layers, more precisely, inequalities between the cross-sections of two layers and their positions as well as atmospheric factors. The full comparison of the mechanical properties’ values for the COLORFILL material in the uniaxial tensile test is presented in Table 10.

The conducted research confirms the results obtained by other researchers, presented by [12,13,14,15], where the authors discussed how the mechanical properties of specimens and components produced by 3D printing change depending on the specimen orientation and filling density specimens, as discussed in this paper. The increase in the level of filling is accompanied by an increase in the value of all tested mechanical quantities (Young’s modulus, tensile strength and yield point). Definite higher values of these parameters are observed for specimens printed in a horizontal orientation. These differences can be clearly seen in the tables presented in Figure 13.

Figure 13 graphically shows the effect of the fill density of specimens printed with COLORFILL filament on the change of mechanical constants, produced in both horizontal (Figure 13a) and vertical (Figure 13b) orientations. In this figure, a trend line drawn as a linear function is drawn for the change of each of the quantities. As can be seen, in many cases, there is a significant dispersion of the values determined during the experimental tests. In the case of specimens printed in a horizontal orientation (Figure 13a), the smallest dispersion of results is characteristic for Young’s modulus and the largest is characteristic for the yield point. Of course, we evaluate the scatter of results here only qualitatively. In the case of specimens produced by incremental technology in a vertical orientation (Figure 13b), a significant dispersion is observed for each of the three analyzed quantities (Young’s modulus, yield strength and tensile strength). When explaining the scatter of the results, the reasons should be sought in the anisotropy of the material used to produce the COLORFILL filament specimens, the mismatch of the printout parameters (filament temperature, table temperature), the change in ambient conditions and errors made during the uniaxial tensile tests (insufficient mounting of the specimens in the jaws of the traverse, lack of tensile axiality and initial tension during the uniaxial tensile test).

Figure 13c,d show the change in the averaged values of the determined material constants for the COLORFILL filament, respectively, when using specimens printed in horizontal and vertical orientations. In the case of plotting specimens printed in a vertical orientation (Figure 13d), when assessing the effect of the filling density on the value of the yield point, the data set is limited only to the results for 60% and 90% fillings, with the non-matching result rejected in the latter case to the observed trend. Figure 13c,d also draw a trend line, determined by the method of least squares, looking for the best fit to the equation of a straight line. In the case of specimens printed in a horizontal orientation (Figure 13c), the fit of a simple regression to the dependence of the Young’s modulus value as a function of the filling density of the specimens, expressed as a percentage, is ideal, similarly in the case of yield strength and tensile strength (in both). In some cases, the level of fit expressed by the coefficient of determination R^2^ exceeds the value of 0.95. Slightly different conclusions should be drawn when analyzing the data for specimens printed in the vertical orientation (Figure 13d). While the fitting of a simple regression to the experimental results for the tensile strength evaluated as a function of the fill density of the vertically printed specimens is characterized by the coefficient of determination R^2^ = 0.99, in the case of Young’s modulus such an obvious dependence of the Young’s modulus increase with the increase in fill density is not observed. The lack of dependence can be seen in the reasons already mentioned above. For this group of specimens, when assessing the impact of filling density on the value of the yield point, the authors do not have the values for the 30% filling, which could not be determined from the obtained σ = f(ε) graphs.

## 4. Discussion on Results

Based on the obtained results of the laboratory tests, the tested material generally shows good mechanical properties in relation to other plastics, for example in relation to PE-HD or ABS materials. It can be boldly suggested that the filament with the trade name COLORFILL can be considered a construction material to be used in the construction of light-weight UAVs. As shown, the material has a relatively high linear stiffness coefficient, which can generally be averaged to 1.71 GPa by analyzing all specimens. The material selected on a UAV is highly anisotropic, which means that its properties in the X and Y axes deviate from the properties in the Z axis. It is therefore advantageous to design elements in which the greatest external forces act in the XY plane and the smallest ones act in the Z axis. A 60% filling in a horizontal orientation is characterized by the best ratio of weight (and with this time for making the detail using the incremental technique) to mechanical properties. The material prepared during 3D printing in a horizontal orientation with a 30% filling is characterized by the lowest own weight (while maintaining certain mechanical properties) and can be used for the production of less-loaded structural elements.

When designing elements intended for FDM production, the anisotropy of the material should be taken into account, which means that different fillings should be modeled as separate materials during FEM tests and the degree of filling in relation to the conditions of use and the target stiffness and load capacity of the structure [3,4,5,8] should be selected.

In order to fully understand the behavior of the material, the Poisson’s ratio, bending strength and other mechanical parameters should also be determined experimentally during the course of extended experimental tests. It may be necessary to assess the material’s toughness, which will allow, in extreme situations, to apply the rules of fracture mechanics or even damage to materials, with appropriate formulation of the assumptions and initial conditions of the analysis.

It should be noted that when designing the structure, it is possible to assume the properties of the material that are not determined during the experimental tests but in the form of those given for a spool of filament, which can be found in the literature in tabular form. For the purposes of the completed Master’s thesis, which became the inspiration for the development of this publication, some material constants from the study were selected. The authors recommend that in order to obtain the most constructive values, different fillings should be treated as separate materials. It therefore requires the development of six anisotropic materials. In determining them, the minimum values achieved during the test were adopted, with the omission of specimens significantly different from the average results. The other values were adopted on the basis of tabular values [18]. The specific mass values were averaged on the basis of horizontal and vertical specimens. The shear modulus G (Kirchhoff’s modulus) was determined for the minimum values of the linear elastic modulus E, using the formula:(2)$$G=\frac{E}{2\left(1+\nu \right)}$$
assuming the value of Poisson’s coefficient ν = 0.35, according to the literature [18].

Thanks to this approach, the design project of the UAV drone presented in the thesis [8] was subjected to comprehensive strength analyses, during which the strength of the rotor, joints and other selected elements of the drone’s structure was assessed. The evaluation of the results of the experimental tests for each of the six materials allowed for the development of the tables of mechanical properties of an anisotropic material required by the Solid Works application (each of the fillings tested during experimental tests), which were used to assess the strength of selected drone structural elements, using FEM, which is presented in detail in [8]. An example of the batch table for the Solid Works application is presented below [8].

The applied approach, taking into account the FEM analyses in the design, significantly affects the form of the drone (UAV). The use of many experimental studies based on the metrology of geometric quantities and mechanical properties allows a comprehensive approach to the issue of design, and the adaptation of the obtained experimental results and the results of empirical considerations to FEM allows to optimize the structure of the drone’s frame, reducing its weight and the level of effective stresses responsible for the destruction of the structure in the critical nodes [8], as shown in Figure 14, Figure 15, Figure 16, Figure 17, Figure 18, Figure 19 and Figure 20. In the future, the authors of the paper will present broader considerations concerning the problem discussed in the last paragraph.

The presented FEM simulation results were made in the Solid Works program, using the Simulation add-on [17]. Figure 14 shows a simplified frame model used in the topology analysis to reduce weight by approximately 50%. The weight of the presented solid is 209.1 g.

Figure 15 shows a virtual simplified model of the frame, with applied boundary conditions and all possible loads. As already mentioned, the analysis was carried out in the Solid Works Simulation 2021 add-on. In the case of the frame, the situation of the rotors generating maximum thrust by blocking the movement of the frame at the chassis attachment points was considered. In this case, the frame will be subjected to the greatest stress. The aim of the analysis is to reduce the mass of the solid by about 50%, while maintaining the stiffness of the element. The analysis of the frame began with the simplification of the solid to achieve the most optimal solution and to simplify the finite element mesh. Some of the holes were removed from the body and the vehicle’s tail was thickened, keeping the distance from the vehicle’s skin.

Defining the boundary conditions began by blocking the displacement of the vehicle chassis’ mounting holes (Figure 15). The influence of gravity on the vehicle frame with an acceleration of 9.81 m/s^2^ was defined. The remaining forces acting on the system were defined based on the weights of individual structural elements. The maximum battery weight of 300 g and the maximum thrust of the tail rotor of 4N were assumed. The analysis conditions were defined, with a minimum node thickness of 6 mm, geometry to be preserved, in which solids were created around the mounting holes and some surfaces. Selecting this option was designed to exclude points within the selected areas in order to leave safe areas for unaccounted-for forces, such as a bending force for blocked holes. The option to keep the symmetry of the test results solid was also selected; YZ is the plane of symmetry during the test. The main and auxiliary objectives of the study were defined. The option to reduce weight by 50% was selected as the main objective. The auxiliary purpose was stress constraint, which means that in the finite elements left behind, stresses will not exceed a predetermined limit. However, this option does not give reliable results when the finite element environment exceeds this limit. In this case, the dimensions of the area should be increased and the test repeated. Figure 16 presents the FEM model divided into finite elements, along with the applied boundary conditions and load. In Figure 16, the drone frame mounting is marked in green and the loads resulting from the battery, gyroscope, other structural elements, thrust and gravity are marked in pink (red arrow in the figure).

The material was modeled using the data shown in Table 11 (anisotropic material). The mesh was optimized as to its size, generating it in high quality—a type based on mixed curvature—with 16 Jacobian points in one 2 mm size of finite element. To fill the FEM model, tetrahedral parabolic elements were used, which are defined by four corner nodes, six mid-lateral nodes and six edges (10 nodes in total). These are the basic finite elements for the Solid Works Simulation application when conducting studies using high-quality meshes. The first static study (primary) was carried out on a model consisting of 211,460 finite elements (i.e., 325,211 nodes), while the secondary study, after topological optimization, used an FEM model consisting of 154,628 finite elements (i.e., 249,176 nodes).

The first step of the analysis was to assess the level of effective stresses (calculated according to the Huber–Mises–Hencky hypothesis) and compare them with the yield strength of the material (Figure 17). As can be seen, the designed frame meets the design assumptions; the effective stresses do not exceed the yield point anywhere in the structure. At some points, the value of the effective stress reaches the level equal to half of the yield stress, 0.5 σ_0_.

Therefore, topological optimization was made based on the results obtained in step 1. The performed topology analysis proves that the most optimal solution is to remove the geometry connecting the fasteners (Figure 18).

Based on the performed topology analysis, the material inside the solid was removed from the area at the rear part of the frame (Figure 18 and Figure 19). The obtained mesh was the basis for introducing changes in the structure. As a result of the analysis, they obtained a smoothed mesh, were exported to an *.stl file and then imported to the part model (Figure 19). The above-mentioned boundary conditions and loads were applied, after which the effective stresses were re-estimated according to the Huber–Mises–Hencky hypothesis (Figure 20).

As can be seen, the change of geometry based on the results of the topological analysis did not lead to an increase in the effective stresses in the considered structure. It should be emphasized that, in the model proposed after topological optimization, the maximum effective stresses decreased by almost 40% compared with the originally estimated ones, while reducing the material necessary to make the frame. In conclusion, it should be noted that the optimized model of the drone frame, with the assumed boundary conditions and loads, showed effective stresses at the level of 30% of the yield strength, which gave a safety factor of three, characteristic for structural steels for various purposes.

A series of FEM calculations, together with topological optimization, allowed to reduce the weight of the drone frame; as well, the introduced changes in geometry led to a reduction of the maximum effective stresses in the nerve center of the structure by almost 38% compared with the original value.

The authors are currently working on a publication that will follow these communications. In the future, the aspect of filament selection will be discussed in the new manuscript, as well as the method of modeling and optimization of the full drone structure (UAV).

## 5. Summary

The paper briefly presents a drone UAV model designed with the use of product synthesis schemes that is intended to be made entirely of components obtained by 3D printing. A filament with the trade name COLORFILL was chosen as the construction material. For the purposes of this manuscript, selected mechanical properties of the filament were assessed by conducting a series of uniaxial tensile tests, assessing the impact of the print orientation and the degree of filling of the specimens (30%, 60%, 90% of filling) on these mechanical properties (Young’s modulus, yield stress, ultimate tensile strength). The test results are presented in graphical and tabular form.

The material used in the construction of the drone, in the form of the COLORFILL filament, is characterized by relatively good mechanical properties, which can be successfully used in the construction of unmanned aerial vehicles and other fields of engineering. However, the anisotropic nature of the material should be taken into account in the case of FDM fabrication. Changing the printing direction of an element changes its behavior under load. In the case of horizontally loaded elements, the material behaves elastically and then plastically. In the case of loads carried in vertical directions, the material is brittle elastic.

Specimens printed in a horizontal orientation show significantly higher, on average by 20%, values of the discussed mechanical properties (yield strength and tensile strength) than specimens printed in a vertical orientation.

Differences in the filling of specimens subjected to uniaxial tension between 60% and 90% of filling are small in terms of mechanical properties. Elements printed with 90% fill are slightly more rigid, at the same time being much heavier and requiring a longer production time. The most optimal solution is to print elements with a density of 60% for medium-duty elements. A 90% infill should only be used where structural rigidity is more important than the weight of the element or in the case of elements with very small dimensions.

The optimization of structures manufactured incrementally by the method of geometry topology analysis can be successfully used in the case of these methods of manufacturing. As shown in the paper, a properly designed structure, after optimization, is characterized by a much lower mass than the original design (by at least 20%), as well as a reduction in effective von Mises stresses by about 40%, which, in the final model, reach the level of 30% of yield point, which gives a relative safety factor equal to three, characteristic of structural steels for various purposes.

## Figures and Tables

**Figure 1 materials-16-01194-f001:**
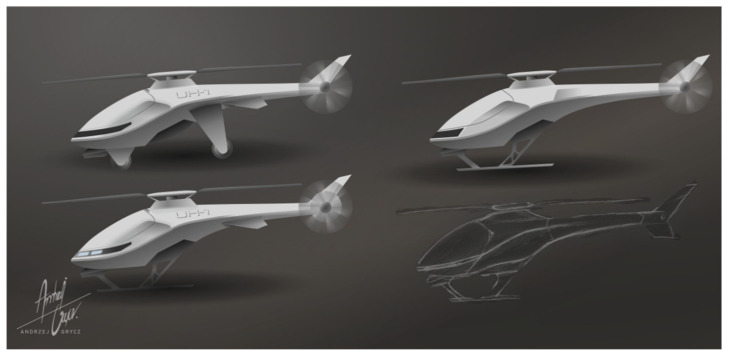
Design variations of the unmanned aerial vehicle (UAV) (drone) (own sketch design, based on [8]).

**Figure 2 materials-16-01194-f002:**
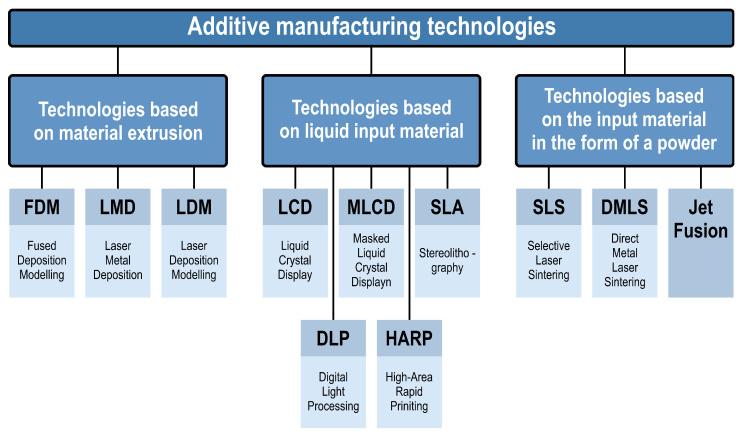
Division of additive manufacturing methods (own elaboration, based on [3,4,5,8]).

**Figure 3 materials-16-01194-f003:**
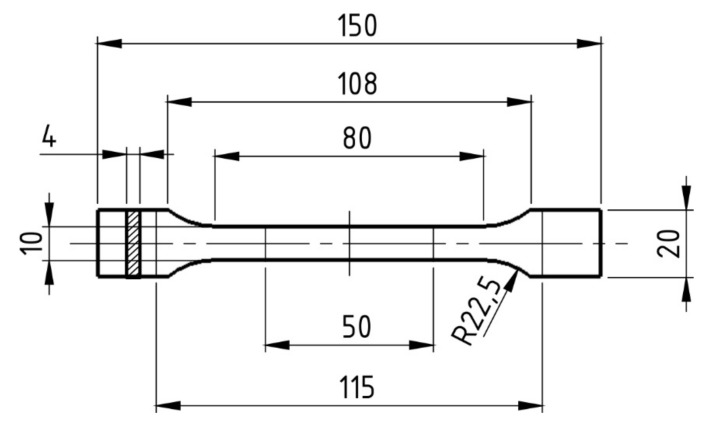
Dimensions of the specimen used in the uniaxial tensile test. All dimensions are given in mm (own elaboration, based on [16]).

**Figure 4 materials-16-01194-f004:**
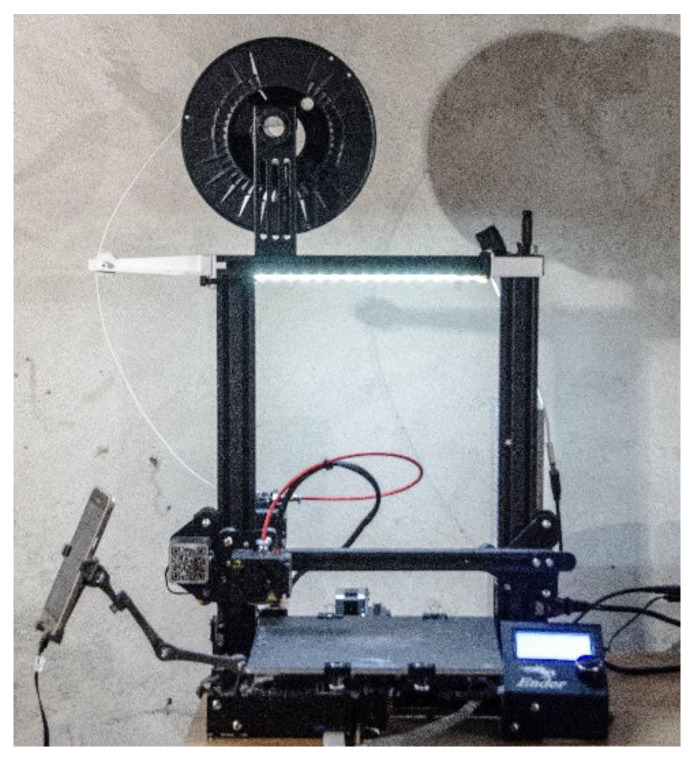
The Creality Ender 3 printer used to produce the specimens.

**Figure 5 materials-16-01194-f005:**
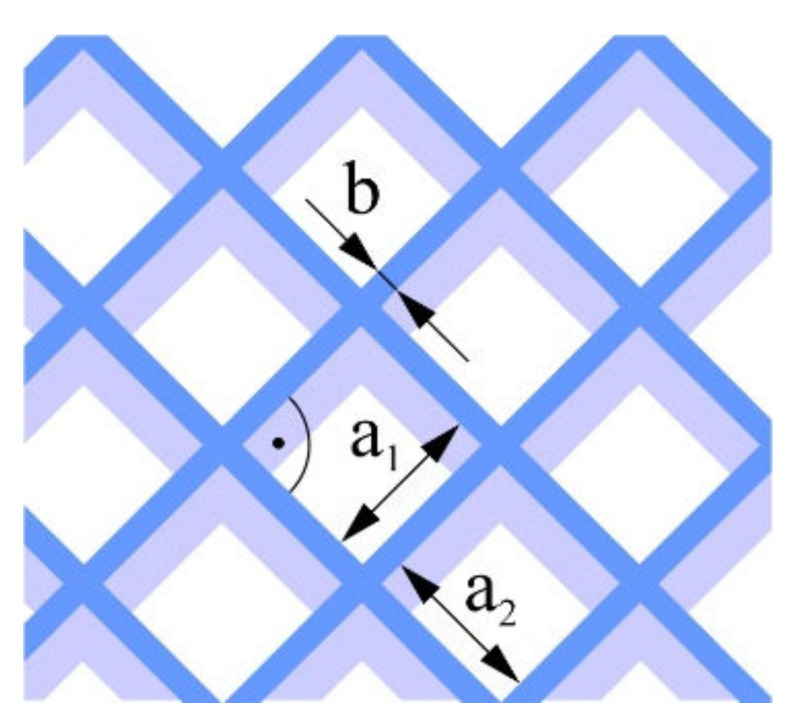
The used structure of the grid filling (b = 0.4mm, a_1_/a_2_ = 1.0).

**Figure 6 materials-16-01194-f006:**
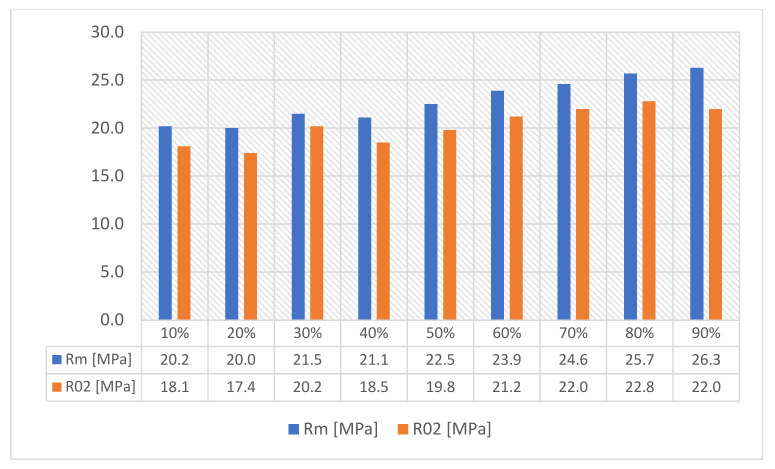
Influence of the density and type of filling for the strength of R_m_ (R^m^) and R_02_ (R02) in the uniaxial tensile test for the type of filling grid (own elaboration, based on [16]).

**Figure 7 materials-16-01194-f007:**
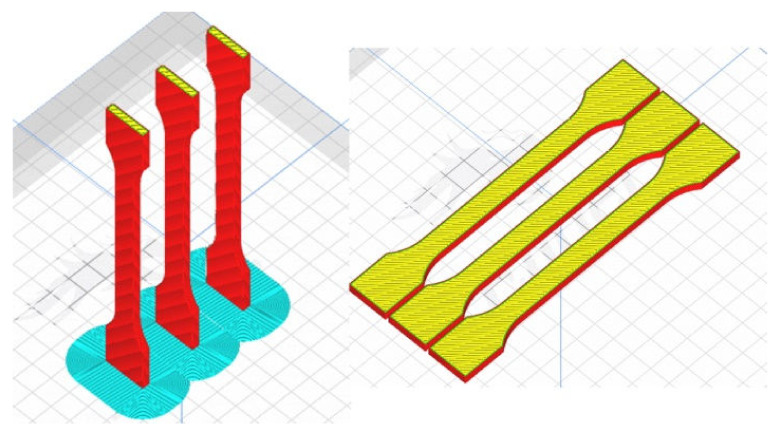
Specimen orientation during printing in Ultimaker Cura.

**Figure 8 materials-16-01194-f008:**
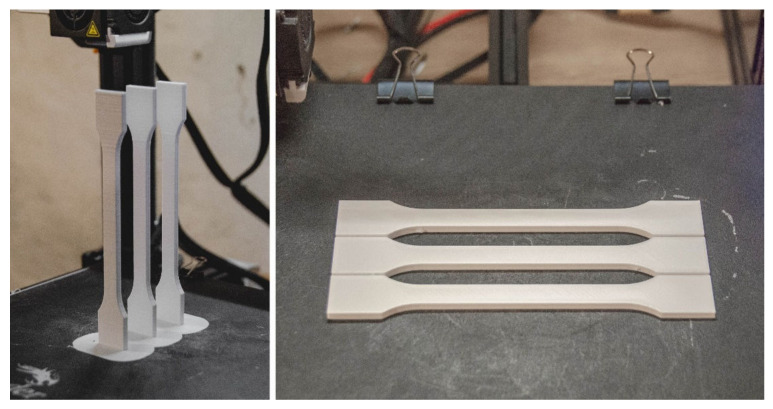
Specimens after the end of the manufacturing process on the worktable.

**Figure 9 materials-16-01194-f009:**
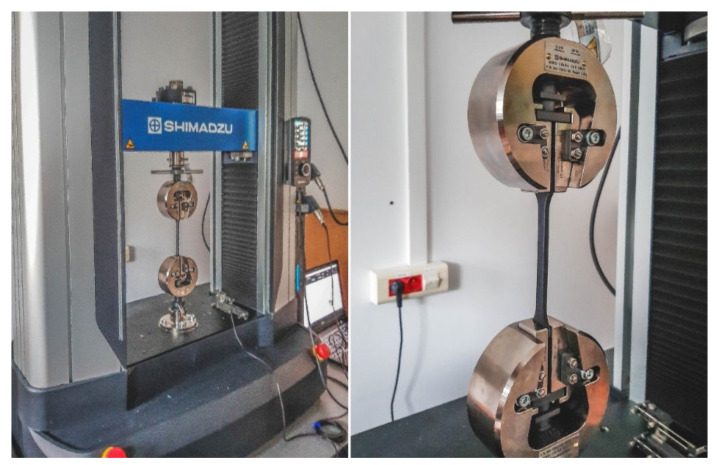
The Shimadzu Autograph AGX-V and the specimen mounted in the jaws.

**Figure 10 materials-16-01194-f010:**
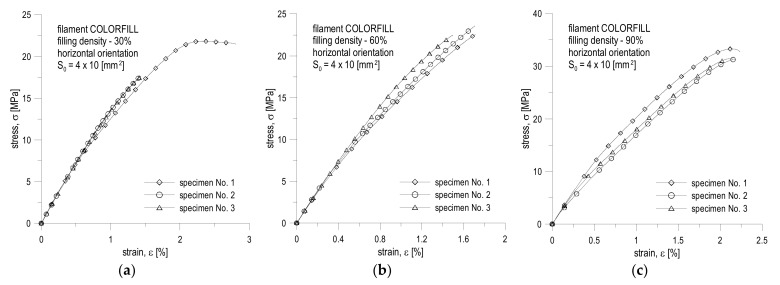
Tension curves for specimens with a different filling for horizontal orientation made of COLORFILL filament: (**a**) 30% filling; (**b**) 60% filling; (**c**) 90% filling.

**Figure 11 materials-16-01194-f011:**
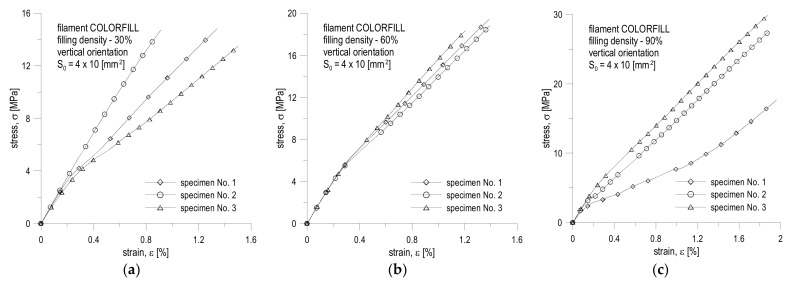
Tension curves for specimens with a different filling for vertical orientation made of COLORFILL filament: (**a**) 30% filling; (**b**) 60% filling; (**c**) 90% filling.

**Figure 12 materials-16-01194-f012:**
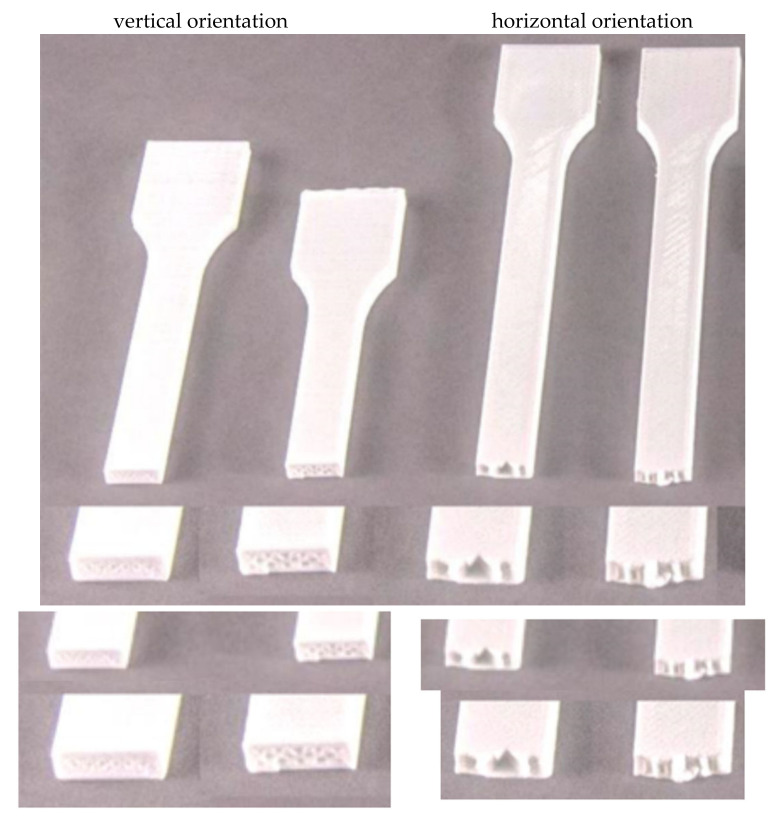
Comparison of the specimen breaking areas for horizontal and vertical orientations.

**Figure 13 materials-16-01194-f013:**
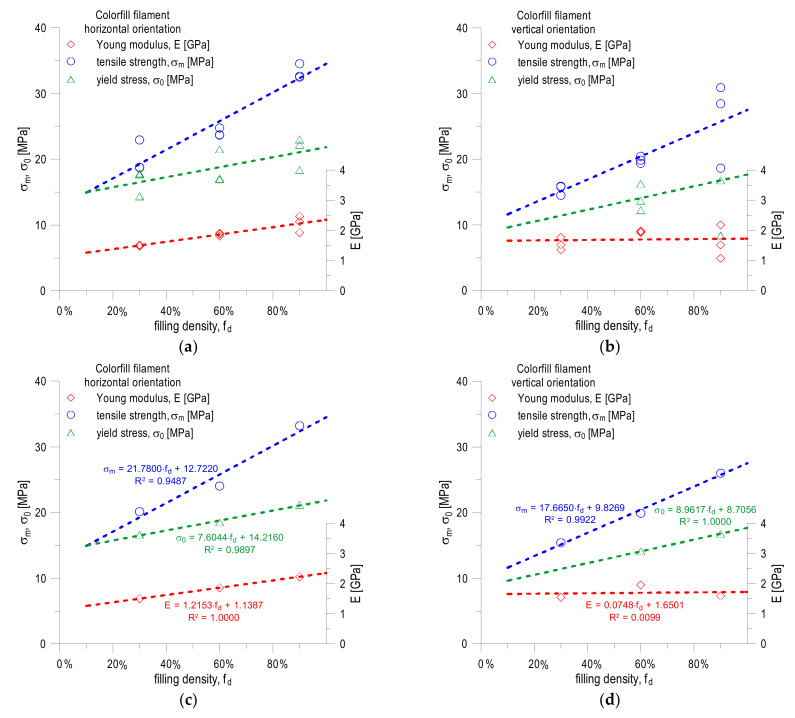
Influence of filling density of specimens printed in horizontal and vertical orientations with COLORFILL filament on mechanical properties (Young’s modulus E, tensile strength σ_m_, yield strength σ_0_): (**a**) comparison of all results obtained in laboratory tests, specimens printed in a horizontal orientation; (**b**) comparison of all results obtained in laboratory tests, specimens printed in a vertical orientation; (**c**) comparison of averaged values, specimens printed in a horizontal orientation; (**d**) comparison of averaged values, specimens printed in a vertical orientation; (**e**–**g**) all obtained results, with error bars calculated as standard deviation.

**Figure 14 materials-16-01194-f014:**
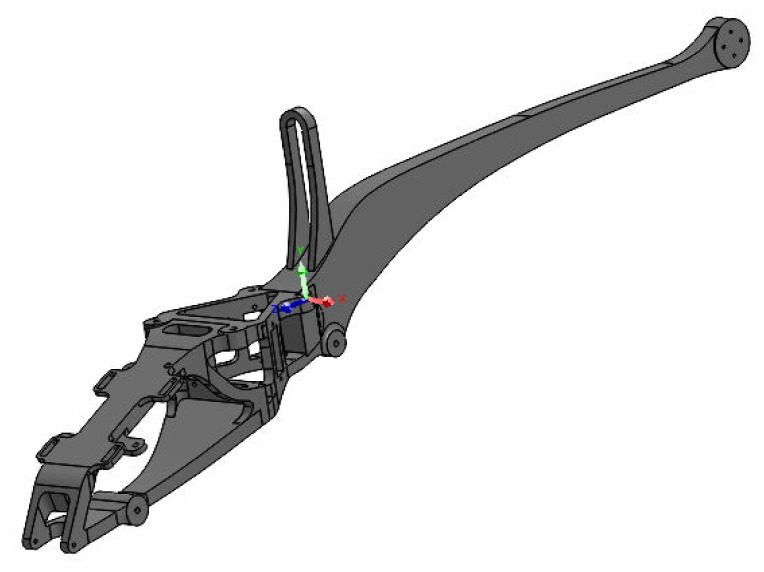
Simplified virtual frame model used in topology analysis of geometry (weight 209.1 g).

**Figure 15 materials-16-01194-f015:**
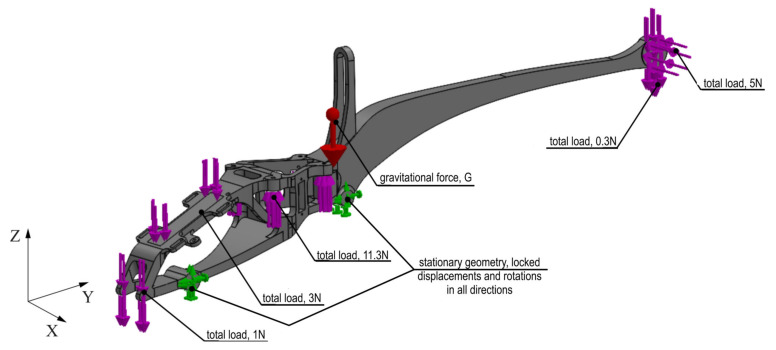
Boundary conditions and loads in frame geometry used in topology analysis.

**Figure 16 materials-16-01194-f016:**
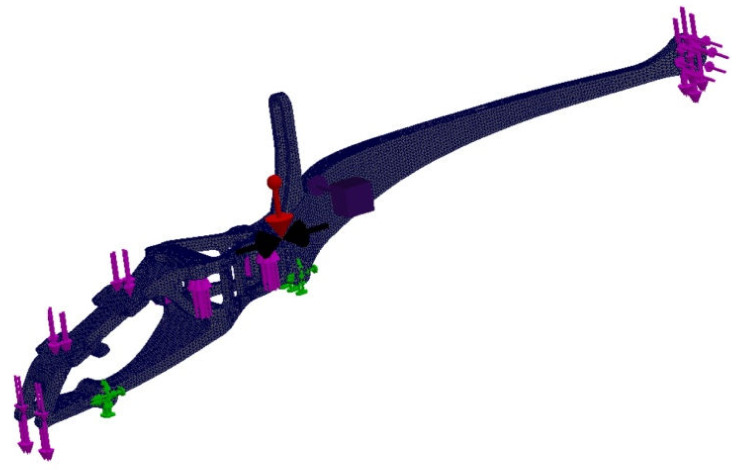
Boundary conditions and loads in frame geometry used in topology analysis, model divided into finite elements.

**Figure 17 materials-16-01194-f017:**
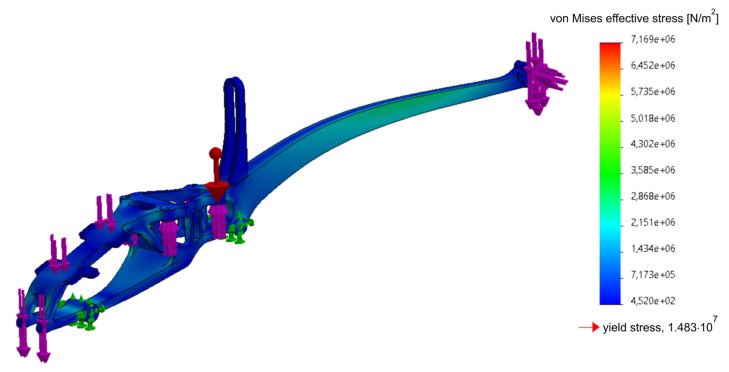
Static analysis of the primary part of the frame (frame weight 180.14 g).

**Figure 18 materials-16-01194-f018:**
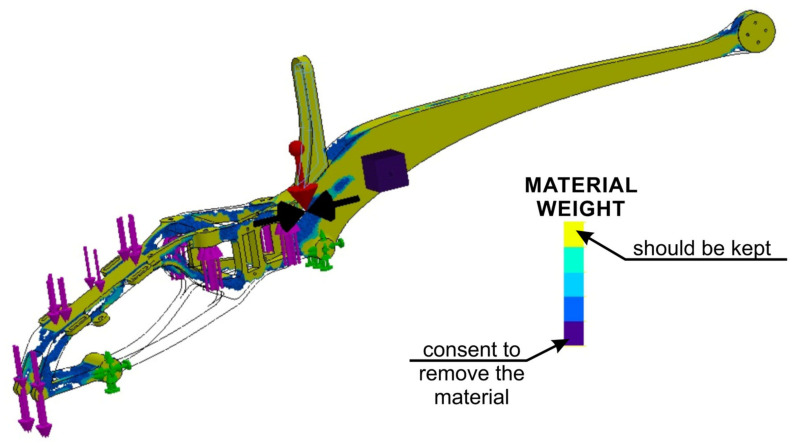
Results of the topological optimization analysis.

**Figure 19 materials-16-01194-f019:**
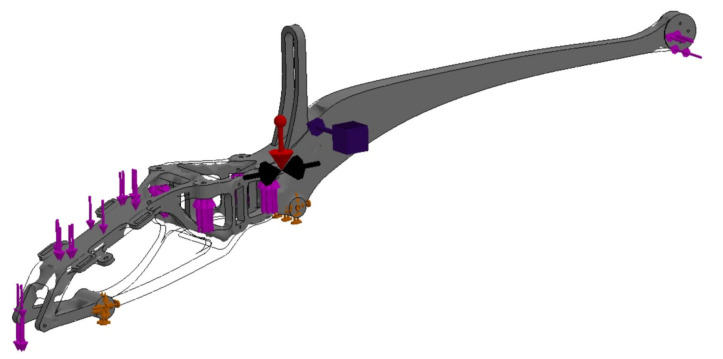
Model of the drone frame with applied boundary conditions and load, prepared based on the results of topological optimization.

**Figure 20 materials-16-01194-f020:**
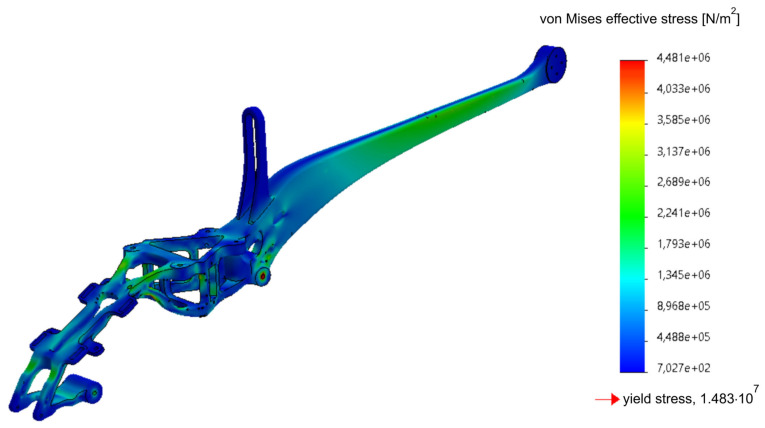
Static analysis of the optimized part of the frame (frame weight 148.34 g).

**Table 1 materials-16-01194-t001:** Constant printing parameters for the Ender 3 printer and Ultimaker Cura.

Parameter Name	Value/ Type	Parameter Name	Value/Type
Head temperature	202 °C	The thickness of the vertical walls	0.8 mm
Table temperature	60 °C	Fill line orientation relative to OX	45°
Layer height	0.2 mm	Fill line width	0.4 mm
Layer width	0.4 mm	Type of support used (horizontal orientation)	No support
Planar head speed on exterior walls	20 mm/s	Type of support used (vertical orientation)	Brim, 15 mm
Planar head speed on the fill	50 mm/s	Layer cooling	From layer No. 4 (0.8 mm)
Thickness of the outer walls	0.8 mm	Worktable	Factory (ready-made)

**Table 2 materials-16-01194-t002:** Comparison of specimen filling in Ultimaker Cura.

Filling Density	Values Approximated by the Program					
	Vertical Orientation (3 Specimens)			Landscape Orientation (3 Specimens)		
	Filling Structure	Printing Time (min)	Used Material (g)	Filling Structure	Printing Time (min)	Used Material (g)
30% $a_{1}=2.(6)\;\left[mm\right]$	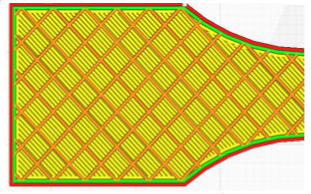	158	18.8	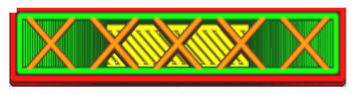	220	20 *
60% $a_{1}=1.(3)\;\left[mm\right]$	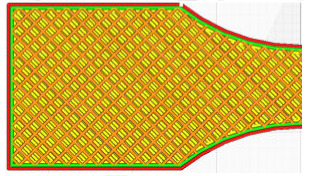	177	23.5	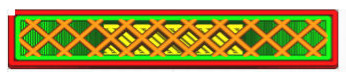	269	24.9 *
90% $a_{1}=0.(8)\;\left[mm\right]$	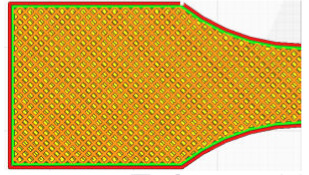	197	28.3	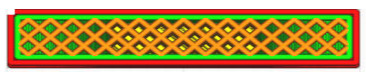	291	30.3 *

* excluding the mass of the support, which in each variant is approximately 0.62 g.

**Table 3 materials-16-01194-t003:** Comparison of specimen weights for COLORFILL PLA material after manufacturing.

Material COLORFILL PLA									
Orientation and Filling	Weight (g)			Total Weight (g)	Weight Value According to the Program (3 Specimens) (g)	Relative Error (3 Specimens) (%)	Weight Value According to the Program (1 Specimen) (g)	Relative Error for Mean Specimen Weight (%)	Average Density of the Specimen Body (kg/m^3^)
	1	2	3						
30% horizontal	5.36	5.26	5.21	15.83	18.8	15.80	6.30	16.24	760.16
60% horizontal	6.58	6.63	6.49	19.7	23.5	16.177	7.80	15.81	946.02
90% horizontal	7.83	7.73	7.78	23.34	28.3	17.53	9.40	17.23	1120.8
30% vertical	6.1	6.07	6.08	18.25	20	8.75	6.60	7.83	876.39
60% vertical	7.56	7.51	7.53	22.6	24.9	9.24	8.20	8.13	1085.29
90% vertical	9.08	9.07	9.06	27.21	30.3	10.20	10.00	9.30	1306.66

**Table 4 materials-16-01194-t004:** Test results for specimens with 30% filling, horizontal orientation, made of COLORFILL filament.

30% Fill Level, Horizontal Orientation COLORFILL	Cross- Sectional Area S (mm^2^)	Maximum Force (N)	Strain for Maximum Force (%)	Linear Elastic Modulus E (GPa)	Tensile Strength R_m_ (MPa)	Yield Stress R_0.2_ (MPa)	Specimen Breaking Energy (J)
Specimen 1	37.64	828.56	2.55	1.52	22.92	14.36	1.93
Specimen 2	37.33	670.52	1.67	1.50	18.72	17.69	0.66
Specimen 3	37.41	670.73	1.70	1.49	18.79	17.85	0.67
Min	37.33	670.52	1.67	1.49	18.72	14.36	0.66
Max	37.64	828.56	2.55	1.52	22.92	17.85	1.93
Average	37.46	723.27	1.97	1.50	20.14	16.63	1.09
Median	37.41	670.73	1.70	1.50	18.79	17.69	0.67

**Table 5 materials-16-01194-t005:** Test results for specimens with a 60% fill, horizontal orientation, made of COLORFILL filament.

60% Fill Level, Horizontal Orientation COLORFILL	Cross- Sectional Area S (mm^2^)	Maximum Force (N)	Strain for Maximum Force (%)	Linear Elastic Modulus E (GPa)	Tensile Strength R_m_ (MPa)	Yield Stress R^0.2^ (MPa)	Specimen Breaking Energy (J)
Specimen 1	35.74	847.1	1.95	1.81	23.70	16.96	0.975
Specimen 2	35.60	1013.0	2.56	1.91	24.74	17.07	2.137
Specimen 3	35.29	835.0	1.91	1.88	23.66	21.51	0.838
Min	35.29	835.0	1.91	1.81	23.66	16.96	0.838
Max	35.74	1013.0	2.56	1.91	24.74	21.51	2.137
Average	35.54	898.4	2.14	1.87	24.04	18.51	1.317
Median	35.60	847.1	1.95	1.88	23.70	17.07	0.975

**Table 6 materials-16-01194-t006:** Test results for specimens with 90% filling, horizontal orientation, made of COLORFILL filament.

90% Fill Level, Horizontal Orientation COLORFILL	Cross-Sectional Area S (mm^2^)	Maximum Force (N)	Strain for Maximum Force (%)	Linear Elastic Modulus E (GPa)	Tensile Strength R_m_ (MPa)	Yield Stress R_0.2_ (MPa)	Specimen Breaking Energy (J)
Specimen 1	35.70	1231.96	2.36	2.47	34.51	22.21	2.005
Specimen 2	35.40	1149.49	2.34	1.93	32.47	23.01	1.676
Specimen 3	36.12	1177.97	2.35	2.29	32.62	18.36	1.742
Min	35.40	1149.49	2.34	1.93	32.47	18.36	1.676
Max	36.12	1231.96	2.36	2.47	34.51	23.01	2.005
Average	35.74	1186.47	2.35	2.23	33.20	21.19	1.808
Median	35.70	1177.97	2.35	2.29	32.62	22.21	1.742

**Table 7 materials-16-01194-t007:** Test results for specimens with 30% filling, vertical orientation, made of COLORFILL filament.

30% Fill Level, Vertical Orientation COLORFILL	Cross- Sectional Area S (mm^2^)	Maximum Force (N)	Strain for Maximum Force (%)	Linear Elastic Modulus E (GPa)	Tensile Strength R_m_ (MPa)	Yield Stress R_0.2_ (MPa)	Specimen Breaking Energy (J)
Specimen 1	39.44	626.54	1.48	1.51	15.88	-	0.548
Specimen 2	39.46	623.96	1.13	1.76	15.81	-	0.363
Specimen 3	40.36	586.03	1.60	1.36	14.50	-	0.582
Min	39.44	586.03	1.13	1.36	14.50	-	0.363
Max	40.36	626.54	1.60	1.76	15.88	-	0.582
Average	39.75	612.17	1.40	1.54	15.40	-	0.498
Median	39.46	623.96	1.48	1.51	15.81	-	0.548

**Table 8 materials-16-01194-t008:** Test results for specimens with 60% filling, vertical orientation, made of COLORFILL filament.

60% Fill Level, Vertical Orientation COLORFILL	Cross- Sectional Area S (mm^2^)	Maximum Force (N)	Strain for Maximum Force (%)	Linear Elastic Modulus E (GPa)	Tensile Strength R_m_ (MPa)	Yield Stress R_0.2_ (MPa)	Specimen Breaking Energy (J)
Specimen 1	41.23	842.97	1.56	1.96	20.45	13.71	0.762
Specimen 2	41.35	868.95	1.67	1.93	19.83	12.26	0.739
Specimen 3	41.51	804.12	1.41	1.97	19.37	16.28	0.627
Min	41.23	804.12	1.41	1.93	19.37	12.26	0.627
Max	41.51	868.95	1.67	1.97	20.45	16.28	0.762
Average	41.36	838.68	1.55	1.95	19.88	14.08	0.709
Median	41.35	842.97	1.56	1.96	19.83	13.71	0.739

**Table 9 materials-16-01194-t009:** Test results for specimens with 90% fill, vertical orientation, made of COLORFILL filament.

90% Fill Level, Vertical Orientation COLORFILL	Cross- Sectional Area S (mm^2^)	Maximum Force (N)	Strain for Maximum Force (%)	Linear Elastic Modulus E (GPa)	Tensile Strength R_m_ (MPa)	Yield Stress R_0.2_ (MPa)	Specimen Breaking Energy (J)
Specimen 1	41.164	766.90	1.65	1.07	18.63	8.40	0.857
Specimen 2	42.164	1200.14	1.90	1.51	28.46	break	1.369
Specimen 3	41.8968	1294.47	2.02	2.18	30.90	16.77	1.534
Min	41.16	766.90	1.65	1.07	18.63	8.40	0.857
Max	42.16	1294.47	2.02	2.18	30.90	16.77	1.534
Average	41.74	1087.17	1.86	1.59	26.00	12.58	1.253
Median	41.90	1200.14	1.90	1.51	28.46	12.58	1.369

**Table 10 materials-16-01194-t010:** Comparison of the mechanical properties’ values for the COLORFILL material in the uniaxial tensile test.

Material Constant	Specimen Orientation	Fill Level	Minimum	Maximum	Average	Difference from the Average Maximum Value for All	Percentage Relative Error in Relation to the Maximum Value (%)
E (GPa)	Horizontal	30%	1.49	1.52	1.50	0.73	32.59
	Vertical	30%	1.36	1.76	1.54	0.69	30.79
	Horizontal	60%	1.81	1.91	1.87	0.36	16.29
	Vertical	60%	1.93	1.97	1.95	0.28	12.41
	Horizontal	90%	1.93	2.47	2.23	Maximum value	Maximum value
	Vertical	90%	1.51	2.18	1.59	0.64	28.85
R_0.2_ (MPa)	Horizontal	30%	14.36	17.85	16.63	4.56	21.52
	Vertical	30%	fault	fault	fault	fault	fault
	Horizontal	60%	16.96	21.51	18.51	2.68	12.65
	Vertical	60%	12.26	16.28	14.08	7.11	33.55
	Horizontal	90%	18.36	23.01	21.19	Maximum value	Maximum value
	Vertical	90%	8.40	16.77	12.59	8.61	40.62
R_m_ (MPa)	Horizontal	30%	18.72	22.92	20.14	13.06	39.33
	Vertical	30%	14.50	15.88	15.40	17.80	53.62
	Horizontal	60%	23.66	24.74	24.03	9.17	27.61
	Vertical	60%	19.37	20.45	19.88	13.32	40.11
	Horizontal	90%	32.47	34.51	33.20	Maximum value	Maximum value
	Vertical	90%	18.63	30.90	26.00	7.20	21.70

**Table 11 materials-16-01194-t011:** Mechanical properties of PLA Colorfill with 30% grid filling [8].

Grid Filling 30% COLORFILL	X/XY	Y/YZ	Z/XZ
Specific mass (kg/m^3^)	818.28		
Modulus of linear elasticity E (GPa)	1.49	1.49	1.36
Yield stress (yield point) R_0.2_ (MPa)	14.36	14.36	break
Yield stress (yield point) R_0.2_ (in FEM tests) (MPa)	14.36		
Tensile strength R_m_ (MPa)	18.72	18.72	14.50
Compressive strength R_s_ (MPa) (43)	93.76		
Poisson’s ratio ν (43)	0.35		
Shear modulus (MPa)	551.85	551.85	503.7

## Data Availability

There is no additional data.

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
