# Peer review of "Assessment of the Mechanical Properties of Selected PLA Filaments Used in the UAV Project"

_materials, 2023, doi:10.3390/ma16031194_

Round 1
Reviewer 1 Report
The manuscript describes the process of mechanical properties evaluation of a PLA filament applied in the 3d printing production of UAV. The manuscript presents some useful information related to the selection of materials for 3d printing, which can be reproduced by the scientific community in this field of research. In general, the manuscript is organized and the results are presented in clear and concise way. However, corrections in several parts of the manuscript are necessary. Also, in my opinion there is no Conclusions section but discussion on results. A new section entitled Conclusions should be added. For detail comments see below:
- Abstract
The abstract should briefly present the results of this work.
- Introduction
Provide number for Introduction.
The Introduction should include a part dealing with related works or literature review. Especially, in relation to the work conducted in this field of study, that is the use of FDM in the production industry and the mechanical properties of the materials applied.
Line 31 ‘…allows you…’ - Rewrite the sentence for clarity.
Line 33-34 ‘…in accordance…’ - Rewrite the sentence for clarity.
Line 69 ‘Figure 2’ - Usually, the full name of FDM is Fused Deposition Modelling. Please argue on changing the term to Fused Distribution Modelling.
Line 71 ‘…FDM technology…’ - Full name should be provided.
In general, in the Introduction section important information is missing. Related works in relation to FDM and mechanical properties of materials applied, gap analysis in the literature, the innovative aspect of this work, aims and objectives, etc.
- 2. Experimental…, conducting research
The last part of the title is not necessary, could be deleted.
Line 96 ‘As can be concluded fin…’ – Check and rewrite the sentence for clarity.
Line 106-11 ‘…from [12] … printed products.’ - Actual results of previous studies mentioned in this paragraph are not discussed. Please be specific and discuss research outcomes from previous works.
Line 133 - Here a new sub-section can be created that discuss the physical tests.
- 3. Measurement results of selected mechanical properties
Line 205-208 ‘It probably resulted…’ - Rewrite the sentence for clarity.
Line 223 ‘Figure 12.’ - Figure 12 is not legible and should be improved.
Line 224 -225 ‘…which were…’ - Rewrite the sentence.
Line 225 – 227 ‘…discussed how…’ - Actual discussion on the results obtained should be provided.
Line 234 – 235 ‘Of course, …’ - This statement should be clarified further.
- 4. Final conclusions
This section should be named Discussion on Results with a subsection at the end about future works.
Line 274 ‘In terms…’ - Rewrite this for clarity. What is the meaning of construction?
Line 278-279 ‘…go from the properties…’ - Rewrite for clarity.
Line 297-298 ‘…in the form…’ - Rewrite for clarity.
Line 322-327 ‘The use of a number…’ - Rewrite the sentence for clarity. Also, it should be shortened.
Line 332 – 334 ‘Contact problems were…’ - Rewrite the sentence for clarity.
Line 351 – 354 ‘The authors are currently…’ - This could be avoided.
A new section entitled Conclusions should be added in the last part of manuscript that concludes the whole work undertaken so far. This is necessary because there is no actual conclusions in the manuscript.
Author Response
Dear Editors, Dear Revierwers,
We would like to thank the Editors of Materials Journal for their valuable suggestions, as well as the three Honorable Reviewers. In the new version of the paper - the corrected version of the manuscript, the guidelines of the Editors as well as three other Honorable Reviewers, to whom we addressed the appropriate answers to all their doubts, were taken into account.
The paper submitted for review has been corrected according to the recommendations of three Honorable Reviewers. We changed the abstract, added keywords, the introduction was changed as suggested, and selected literature items were referred to in more detail.
We improved the quality of the figures, and added other figures to increase the substantive value of the manuscript as suggested by one of the Reviewers. We changed the description of the FEM model and described our simulations in detail. We standardized the notation of numbers in the tables, in accordance with the entry in the text of the paper. In the case of selected figures, this cannot be done, because the meaning of the decimal place depends on the software version available for a given region (for us – Poland).
According to the Reviewers' suggestions, we changed the titles of the sections, corrected their numbering, We added a clear section "5. Conclusions". We have provided accurate conclusions based on our research.
The paper was checked for typos and vocabulary, the figures containing Polish were corrected.
We hope that the paper submitted for re-review meets the requirements of the Editors and Honorable Reviewers and will be qualified for publication in the MDPI Materials Journal.
If there is a need for further corrections, we are ready to introduce them to the paper so that it meets the expectations of the MDPI Journal Editors and Honorable Reviewers.
Sincerely, Marcin Graba, Andrzej Grycz
Reviewer No. 1
Comments and Suggestions for Authors
The manuscript describes the process of mechanical properties evaluation of a PLA filament applied in the 3d printing production of UAV. The manuscript presents some useful information related to the selection of materials for 3d printing, which can be reproduced by the scientific community in this field of research. In general, the manuscript is organized and the results are presented in clear and concise way. However, corrections in several parts of the manuscript are necessary. Also, in my opinion there is no Conclusions section but discussion on results. A new section entitled Conclusions should be added. For detail comments see below:
ANSWER: Thank you for your specific comments and a good word about our pape. The "Summary" section has been added to the manuscript.
- Abstract
The abstract should briefly present the results of this work.
ANSWER: The abstract has been changed.
- Introduction
Provide number for Introduction.
ANSWER: Added number in section title.
The Introduction should include a part dealing with related works or literature review. Especially, in relation to the work conducted in this field of study, that is the use of FDM in the production industry and the mechanical properties of the materials applied.
ANSWER: Corrections have been made in the introduction. By definition, the paper is submitted as "Communication", therefore, according to the authors, there is no need to present such a detailed review of the literature as is the case with original scientific articles or review papers.
Line 31 ‘…allows you…’ - Rewrite the sentence for clarity.
ANSWER: Sentences have been changed in the text.
Line 33-34 ‘…in accordance…’ - Rewrite the sentence for clarity.
ANSWER: Sentences have been changed in the text.
Line 69 ‘Figure 2’ - Usually, the full name of FDM is Fused Deposition Modelling. Please argue on changing the term to Fused Distribution Modelling.
ANSWER: Changes were introduced in the text of the paper and Figure 2 in the manuscript was changed.
Line 71 ‘…FDM technology…’ - Full name should be provided.
ANSWER: Changes have been made to the text of the thesis - the full name has been added.
In general, in the Introduction section important information is missing. Related works in relation to FDM and mechanical properties of materials applied, gap analysis in the literature, the innovative aspect of this work, aims and objectives, etc.
ANSWER: Corrections have been made to the introduction. By design, the paper is submitted as "Communication", so according to the authors, there is no need to present such a detailed review of the literature, as is the case with original scientific articles or review papers.
- 2. Experimental…, conducting research
The last part of the title is not necessary, could be deleted.
ANSWER: The section title has been corrected.
Line 96 ‘As can be concluded fin…’ – Check and rewrite the sentence for clarity.
ANSWER: The text in the paper has been corrected and changed. Figure 6 in the paper was made by the authors on the basis of source data.
Line 106-11 ‘…from [12] … printed products.’ - Actual results of previous studies mentioned in this paragraph are not discussed. Please be specific and discuss research outcomes from previous works.
ANSWER: The authors, as far as possible, made changes in the fragment being the subject of discussion. The text in lines 106-11 refers to several papers that the authors became acquainted with while working on the presented material. In this part of the section, the authors try to present what other researchers are doing in the field of design and use in prototyping 3D printing. Indeed, the actual results of these studies are not discussed here, however, the authors suggest the possibility of finding the literature by readers. The section on experimental research is, in the authors' opinion, not suitable for reviewing other papers. The current paper is a typical "Communication" - it is supposed to bring the reader closer to the authors' activities, and indicates that in the future the authors will devote more attention to certain issues, including the discussion of the results of other authors' research.
Line 133 - Here a new sub-section can be created that discuss the physical tests.
ANSWER: Thank you for your attention, however, we will not divide this section - the preparation of specimens, carrying out tensile tests, measurements of mass and geometric dimensions are quite closely related to each other.
- 3. Measurement results of selected mechanical properties
Line 205-208 ‘It probably resulted…’ - Rewrite the sentence for clarity.
ANSWER: The sentence has been corrected.
Line 223 ‘Figure 12.’ - Figure 12 is not legible and should be improved.
ANSWER: Changed image resolution, added close-ups of sample cracks.
Line 224 -225 ‘…which were…’ - Rewrite the sentence.
ANSWER: The sentence has been corrected.
Line 225 – 227 ‘…discussed how…’ - Actual discussion on the results obtained should be provided.
ANSWER: Added description of actual results.
Line 234 – 235 ‘Of course, …’ - This statement should be clarified further.
ANSWER: The text of the paper has been corrected. The presented statements have been clarified.
- 4. Final conclusions
This section should be named Discussion on Results with a subsection at the end about future works.
ANSWER: The title of the section has been changed, as suggested by the Honorable Reviewer.
Line 274 ‘In terms…’ - Rewrite this for clarity. What is the meaning of construction?
ANSWER: The text of the manuscript has been corrected.
Line 278-279 ‘…go from the properties…’ - Rewrite for clarity.
ANSWER: The text of the paper has been corrected.
Line 297-298 ‘…in the form…’ - Rewrite for clarity.
ANSWER: The sentence has been changed.
Line 322-327 ‘The use of a number…’ - Rewrite the sentence for clarity. Also, it should be shortened.
ANSWER: The sentence has been corrected as suggested.
Line 332 – 334 ‘Contact problems were…’ - Rewrite the sentence for clarity.
ANSWER: The text of the paper has been corrected. The description of the FEM model has been changed.
Line 351 – 354 ‘The authors are currently…’ - This could be avoided.
ANSWER: The text of the paper has been corrected as suggested by the Honorable Reviewer and the other two Reviewers.
A new section entitled Conclusions should be added in the last part of manuscript that concludes the whole work undertaken so far. This is necessary because there is no actual conclusions in the manuscript
ANSWER: The Conclusions section has been added and supplemented according to the suggestions of all Reviewers.
Reviewer 2 Report
The paper deals with mechanical properties of PLA filaments used for UAV production. A good methodology was applied in this study as well as discussing the current technology. I suggest publication after some revisions given below.
· Please check the text for typos and grammar.
· Please clearly state the novelty in this work.
· Better to give discuss the previous works individually in Introduction instead of giving bulk citations.
· Please give a statement as “dimensions in mm” in fig.3.
· I think fig.4 should be removed.
· Is there a design methodology in the specimen design such as Taguchi approach or any other one?
· In the tables, digits are separated by commas however by dots in the text. Please use a constant separator.
· Charts such as in fig.10 should be bigger to clearly seen by the readers.
· What about the deviations in each test? Authors may use error bars to indicate it.
· The heading “4. Final conclusions” should be changed by a proper one.
· The details of the FEM should be extended and given in detail.
· Fracture surfaces can be analyzed to have an idea about the mechanical properties.
Author Response
Dear Editors, Dear Revierwers,
We would like to thank the Editors of Materials Journal for their valuable suggestions, as well as the three Honorable Reviewers. In the new version of the paper - the corrected version of the manuscript, the guidelines of the Editors as well as three other Honorable Reviewers, to whom we addressed the appropriate answers to all their doubts, were taken into account.
The paper submitted for review has been corrected according to the recommendations of three Honorable Reviewers. We changed the abstract, added keywords, the introduction was changed as suggested, and selected literature items were referred to in more detail.
We improved the quality of the figures, and added other figures to increase the substantive value of the manuscript as suggested by one of the Reviewers. We changed the description of the FEM model and described our simulations in detail. We standardized the notation of numbers in the tables, in accordance with the entry in the text of the paper. In the case of selected figures, this cannot be done, because the meaning of the decimal place depends on the software version available for a given region (for us – Poland).
According to the Reviewers' suggestions, we changed the titles of the sections, corrected their numbering, We added a clear section "5. Conclusions". We have provided accurate conclusions based on our research.
The paper was checked for typos and vocabulary, the figures containing Polish were corrected.
We hope that the paper submitted for re-review meets the requirements of the Editors and Honorable Reviewers and will be qualified for publication in the MDPI Materials Journal.
If there is a need for further corrections, we are ready to introduce them to the paper so that it meets the expectations of the MDPI Journal Editors and Honorable Reviewers.
Sincerely, Marcin Graba, Andrzej Grycz
Reviewer No. 2
Comments and Suggestions for Authors
The paper deals with mechanical properties of PLA filaments used for UAV production. A good methodology was applied in this study as well as discussing the current technology. I suggest publication after some revisions given below.
- Please check the text for typos and grammar.
ANSWER: The article has been checked for typos as well as grammar.
- Please clearly state the novelty in this work.
ANSWER: The manuscript was corrected as suggested by the Honorable Reviewer. Abstact has been changed, conclusions have been added, the sense of experimental work and simulations has been emphasized.
- Better to give discuss the previous works individually in Introduction instead of giving bulk citations.
ANSWER: Corrections have been applied as suggested by the Honorable Reviewer. Changes were made in the introduction, referring to literature items [12-15], a short extension of the text was made, mentioning the results and conclusions contained in these manuscripts.
- Please give a statement as “dimensions in mm” in fig.3.
ANSWER: In accordance with the applicable ISO rules in the technical drawing, we do not provide units on the dimension lines - the unit was inserted in the figure description - [mm].
- I think fig.4 should be removed.
ANSWER: Analyzing the paper and the received three reviews, as well as the editor's comments, it was decided to leave this figure in the paper.
- Is there a design methodology in the specimen design such as Taguchi approach or any other one?
ANSWER: We know this method, but we do not focus directly on the quality of our solution - the drone product. Referring to the Taguchi method would require a slightly different paper profile from us. We try to focus on the mechanical properties and strength of the structure. However, as the Honorable Reviewer pointed out, this is a very important problem, worthy of attention and discussion and indicating the need to use the Taguchi Method in design and engineering, due to the growing competition and globalization. We do not exclude a reference to this method in our next paper.
- In the tables, digits are separated by commas however by dots in the text. Please use a constant separator.
ANSWER: The tables have been formatted as numbers in the text of the thesis. Unfortunately, we have no influence on the format of numbers in some figures - it was developed by developers of computer programs for our region, which is Poland.
- Charts such as in fig.10 should be bigger to clearly seen by the readers.
ANSWER: The figures have been enlarged, if the Publisher decides, they will be enlarged even more.
- What about the deviations in each test? Authors may use error bars to indicate it.
ANSWER: The deviation can be estimated individually by the reader. Graphs with error bars have been added, as suggested by the Honorable Reviewer.
- The heading “4. Final conclusions” should be changed by a proper one.
ANSWER: Added section "5. Conclusions", retitled section 4.
- The details of the FEM should be extended and given in detail.
ANSWER: Sections on FEM modeling have been expanded.
- Fracture surfaces can be analyzed to have an idea about the mechanical properties.
ANSWER: The paper is not about cracks. This is indicated in the text of the paper.

Reviewer 3 Report
This manuscript presented the experiments for analyzing the mechanical properties of PLA filaments printed in different orientations and 3 filling settings. And the authors claimed that this manuscript came from the work for author’s master thesis.
There was no novelty in the repeating the uniaxial tensile test for PLA filament, as dozens of papers related to this topic have been published since 2010. And authors should emphasize the novelty of this communication clearly. The manuscript title was “Assessment of the mechanical properties of selected PLA filaments used in the UAV project”, however, the introduction focused on the description of UAV design and additive manufacturing methods. The introduction was poorly written and confusing. There are a few suggests or questions as follows:
1. In the introduction part, the reasons why FDM method and PLA filaments were chosen in this study were not clearly clarified, and relevant citations of other researchers’ work which included assessment of mechanical properties of 3D-printed specimens or 3D-printed UAV, were missing. 1/3 of the references came from the authors’ master thesis or books written in Polish. I don’t think the authors organized the references properly.
2.Page 5, Line 119, specimens printed in vertical and horizontal orientation were adopted and used material in landscape orientation excluded the mass of the support. This paper utilized the standard specimen structure for PLA test, but when facing UAV design, complex structures were often considered and here support was needed. The influence on the mechanical properties of PLA filaments, after introducing necessary support, should be discussed.
3.The critical flaw in this manuscript was the test results and test analysis for specimens with different fillings. In each group, only 3 repeated testing was considered. Some unnormal results appeared and were ignored by the authors. For example, the results for specimen 1 in Table 4 or specimen 2 in Table 5. Others were not listed and authors should check these carefully. These experiment results might affect the validity of conclusions.
4.Page 15, language in Figure 14&15 should be corrected in English.
Author Response
Dear Editors, Dear Revierwers,
We would like to thank the Editors of Materials Journal for their valuable suggestions, as well as the three Honorable Reviewers. In the new version of the paper - the corrected version of the manuscript, the guidelines of the Editors as well as three other Honorable Reviewers, to whom we addressed the appropriate answers to all their doubts, were taken into account.
The paper submitted for review has been corrected according to the recommendations of three Honorable Reviewers. We changed the abstract, added keywords, the introduction was changed as suggested, and selected literature items were referred to in more detail.
We improved the quality of the figures, and added other figures to increase the substantive value of the manuscript as suggested by one of the Reviewers. We changed the description of the FEM model and described our simulations in detail. We standardized the notation of numbers in the tables, in accordance with the entry in the text of the paper. In the case of selected figures, this cannot be done, because the meaning of the decimal place depends on the software version available for a given region (for us – Poland).
According to the Reviewers' suggestions, we changed the titles of the sections, corrected their numbering, We added a clear section "5. Conclusions". We have provided accurate conclusions based on our research.
The paper was checked for typos and vocabulary, the figures containing Polish were corrected.
We hope that the paper submitted for re-review meets the requirements of the Editors and Honorable Reviewers and will be qualified for publication in the MDPI Materials Journal.
If there is a need for further corrections, we are ready to introduce them to the paper so that it meets the expectations of the MDPI Journal Editors and Honorable Reviewers.
Sincerely, Marcin Graba, Andrzej Grycz
Reviewer No. 3
Comments and Suggestions for Authors
This manuscript presented the experiments for analyzing the mechanical properties of PLA filaments printed in different orientations and 3 filling settings. And the authors claimed that this manuscript came from the work for author’s master thesis.
ANSWER: The paper was based on the master's thesis of one of the authors. The idea has been developed, research is continued and in the future it will certainly result in another publication on 3D printing, the strength of additively produced structural elements, and computer simulations.
There was no novelty in the repeating the uniaxial tensile test for PLA filament, as dozens of papers related to this topic have been published since 2010. And authors should emphasize the novelty of this communication clearly. The manuscript title was “Assessment of the mechanical properties of selected PLA filaments used in the UAV project”, however, the introduction focused on the description of UAV design and additive manufacturing methods. The introduction was poorly written and confusing. There are a few suggests or questions as follows:
ANSWER: The tests carried out for the purposes of the thesis and master's thesis of one of the authors are not new. The authors got acquainted with many literature items, the abstract and the introduction were changed, many fragments of the paper were extended, section "5. Conclusions" was added. Corrections were made according to the suggestions of all three Honorable Reviewers. This paper is treated as "Communication". Our paper presents groundbreaking preliminary results and significant findings that are part of a larger study over multiple years. It also include cutting-edge methods and experiments, and the development of new technology and materials.
- In the introduction part, the reasons why FDM method and PLA filaments were chosen in this study were not clearly clarified, and relevant citations of other researchers’ work which included assessment of mechanical properties of 3D-printed specimens or 3D-printed UAV, were missing. 1/3 of the references came from the authors’ master thesis or books written in Polish. I don’t think the authors organized the references properly.
ANSWER: Literature was checked, reference was made to several literature items in the text of the paper, as suggested by the Honorable Reviewer.
2.Page 5, Line 119, specimens printed in vertical and horizontal orientation were adopted and used material in landscape orientation excluded the mass of the support. This paper utilized the standard specimen structure for PLA test, but when facing UAV design, complex structures were often considered and here support was needed. The influence on the mechanical properties of PLA filaments, after introducing necessary support, should be discussed.
ANSWER: In the case of both groups of specimens, no support material was used - there was no such need. Yes, we agree with the comment that often supports are necessary in additive manufacturing. In the case of uniaxial tensile tests - specimens can be prepared without supports - unless an unnatural arrangement of the specimen on the work table is expected - then a support is necessary. It is possible that the authors will be tempted to examine the impact of the unnatural arrangement of the specimen on the work table, taking into account the supports.
3.The critical flaw in this manuscript was the test results and test analysis for specimens with different fillings. In each group, only 3 repeated testing was considered. Some unnormal results appeared and were ignored by the authors. For example, the results for specimen 1 in Table 4 or specimen 2 in Table 5. Others were not listed and authors should check these carefully. These experiment results might affect the validity of conclusions.
ANSWER: The authors are aware that the results of the tests carried out are not fully reproducible. The paper presents all the results obtained, without "stretching" the results to present the problems arising during the experimental work. The number of specimens resulted from the limited budget that the authors had at their disposal during the implementation of research work and conducting all analyzes and simulations. The authors planned a larger research project that, with funding, would allow for the assessment of more mechanical properties, in populations with much more specimens and combinations of other 3D printing parameters. Many publications, including those from the MDPI journal group, are based on research conducted in a much smaller population. The authors mentioned any disturbances in the results in the paper, also correcting the text of the manuscipt, in accordance with the comments of the Honorable Reviewer.
4.Page 15, language in Figure 14&15 should be corrected in English.
ANSWER: The text in Polish in the figures has been changed to English, as suggested by the Honorable Reviewer.

Round 2
Reviewer 2 Report
Acceptable